# Beyond Retrieval Competition: Asymmetric Effects of Retroactive and Proactive Interference in Associative Memory

**DOI:** 10.3390/bs15111459

**Published:** 2025-10-27

**Authors:** Yahui Zhang, Weihai Tang, Mei Peng, Xiping Liu

**Affiliations:** 1Faculty of Psychology, Tianjin Normal University, Tianjin 300387, China; 2Development of Psychology, University of Sanya, Sanya 572022, China; 3School of Mental Health, Wenzhou Medical University, Wenzhou 325035, China

**Keywords:** proactive interference, retroactive interference, associative recognition, source memory, encoding processes, response time

## Abstract

Although associative interference has traditionally been attributed to retrieval competition, emerging evidence suggests that interference may also arise from encoding-based representational processes. The present study examined whether retroactive interference (RI) and proactive interference (PI) can occur in the absence of explicit retrieval competition and whether they reflect distinct underlying mechanisms. Participants studied two lists of word–picture pairs in an AB/AC associative learning paradigm, followed by a non-competitive two-alternative forced-choice (2AFC) associative recognition test and a source memory task. Across both frequentist and Bayesian analyses, recognition accuracy revealed a significant RI effect—lower accuracy for earlier A-B pairs relative to non-overlapping controls—whereas PI manifested as longer reaction times (RTs) for later A-C pairs, despite comparable accuracy. Source judgments showed faster correct responses for overlapping than for non-overlapping pairs, suggesting that cue overlap facilitated more fluent retrieval rather than confusion. These findings indicate that interference can emerge independently of retrieval competition and that RI and PI are supported by dissociable mechanisms: RI reflects encoding-related reorganization that weakens earlier associations, whereas PI reflects increased retrieval effort following differentiation of overlapping traces. Together, the results support a process-interaction framework in which encoding-based reactivation and reorganization shape later retrieval dynamics, demonstrating that associative interference arises from the interplay between encoding and retrieval processes rather than retrieval competition alone.

## 1. Introduction

### 1.1. Associative Interference in Episodic Memory

Episodic memory is defined by its rich network of associative connections that link elements co-occurring within a specific spatial and temporal context. These associative bindings—such as the connection between a person’s face and their name—enable individuals to recall not only isolated items but also the relationships among them. Through such associative structures, episodic memory supports flexible retrieval and promotes generalization across related experiences ([19]; [28]). However, because different episodes often share overlapping elements, these shared components can also become sources of interference when they serve as retrieval cues ([10]). As a result, the very mechanism that enables memory integration and generalization can also introduce competition among related traces, sometimes impairing accurate retrieval.

This is particularly relevant in the AB/AC paradigm, also known as the paired associate learning paradigm ([14]; [24]), which has often been used to investigate associative interference. In this paradigm, participants study two lists of cue-target pairs (e.g., picture-word pairs). In List 1, each cue (A) is paired with a unique target (B), forming A-B pairs. In List 2, the same cues are re-paired with new targets (C), forming A-C pairs. Retroactive interference (RI) is typically assessed by comparing memory for the A-B pairs between participants who do or do not study the second list (A-C pairs), whereas proactive interference (PI) is measured by comparing memory for the A-C pairs between participants who have or have not studied the first list (A-B pairs). These traditional “pure” control designs, in which one list is omitted, help isolate interference caused by competing associations ([14]; [25]).

However, such between-group control procedures have two major limitations. First, they lack ecological validity—real-world memory rarely involves conditions where prior or subsequent learning is entirely absent. Second, they fail to disentangle interference caused by cue-based associative overlap from more general effects of list context or cognitive load.

To overcome these limitations, the present study adopted a within-subject, similarity-based contrast design. All participants studied both lists, but only some pairs involved overlapping cues (A-B/A-C), whereas others consisted of entirely new cue–target combinations (E-F/G-H). The non-overlapping pairs served as an internal control condition, allowing interference effects to be attributed specifically to cue overlap and associative similarity, rather than to overall task demands or list order. This mixed-list paradigm provides a more precise and ecologically valid means of examining interference and integration processes, and follows recent advances emphasizing representational overlap as a critical determinant of memory interaction ([4]; [12]; [29]).

### 1.2. Two-Factor Interference Theory: Retrieval Competition and Associative Unlearning

Classic theories of interference attribute memory disruption primarily to retrieval-phase processes. The retrieval competition account ([15]; [37]) posits that cue overload diminishes retrieval specificity, leading to competition among similar memories.

The Search of Associative Memory (SAM) model ([16]; [17]; [27]) formalizes this process, representing memory as a network of interconnected nodes, where a cue activates multiple nodes, and their relative activation strengths determine retrieval success. In the SAM model, retrieval is governed by a probabilistic process where the probability of retrieving a specific memory trace is proportional to its activation strength divided by the sum of the activation strengths of all traces associated with the cue. This competitive process means that stronger traces dominate retrieval, reducing the likelihood of retrieving weaker traces and causing interference ([6]; [8]; [31]).

In contrast to retrieval-based accounts, the associative unlearning hypothesis proposes that retroactive interference arises during encoding rather than retrieval. Specifically, learning cue-overlapping associations in List 2 is thought to weaken previously acquired associations in List 1 ([2]). This weakening is particularly evident when cue–outcome contingencies are violated, disrupting the stability of existing associative links ([20]). However, most evidence for associative unlearning has come from overlearning paradigms, and its relevance to typical episodic memory situations remains uncertain.

### 1.3. Dynamic Interactions Between Encoding and Retrieval in Memory Interference

Recent research suggests that when people learn new overlapping associations (A-C), the process can trigger reactivation of earlier related memories (A-B), leading the two associations to become either integrated or differentiated ([4]; [12], [11]; [29]). Such similarity-based reactivation engages hippocampal mechanisms that shape how old and new memories interact during encoding ([40], [41]; [30]). Strong reactivation favors pattern completion, promoting integration and the formation of cohesive memory networks, whereas moderate reactivation supports pattern separation, enhancing differentiation and reducing interference. When reactivation is weak or absent, encoding proceeds relatively independently, leaving both memories unchanged. Such conditions minimize interference but also prevent beneficial reorganization processes such as integration or differentiation ([21]; [32]). These encoding-based processes determine the extent to which overlapping memories share or segregate their representational structure, influencing their later accessibility.

Overall, cue overlap amplifies these encoding-based dynamics by increasing the likelihood of reactivating prior associations, thereby heightening interference effects for overlapping (A-B/A-C) pairs compared to non-overlapping (E-F/G-H) pairs. Neural evidence further suggests that hippocampal pattern separation supports differentiation, whereas medial prefrontal cortex (mPFC) activity promotes integration across related experiences ([32]). These findings indicate that both retroactive and proactive interference may originate not solely from retrieval competition or unlearning, but from representational reorganization during encoding, where the degree of reactivation determines whether memories become integrated or differentiated.

Crucially, the retrieval competition framework provides a complementary perspective on how such reorganized representations manifest during memory access. According to this view, interference arises when multiple associations linked to the same cue become coactivated at retrieval, leading to competition between the target and non-target traces ([1]; [16]; [27]). The degree of competition depends on the overlap and distinctiveness of the underlying representations—features that are themselves shaped by encoding dynamics. Specifically, integration increases associative overlap, heightening cue-driven coactivation and thereby intensifying retrieval competition. In contrast, differentiation reduces shared features, facilitating selective retrieval of the target association and mitigating interference.

From this integrated perspective, encoding-based reactivation and retrieval-based competition are not independent phenomena but complementary stages of the same process. Encoding determines the structural similarity among memory traces, while retrieval reveals the functional consequences of that structure. Thus, the extent of proactive and retroactive interference observed during associative and source recognition tasks reflects the interaction between how overlapping memories are reorganized at encoding and how they compete for activation at retrieval. This framework underscores that memory interference emerges from dynamic interactions across both stages, rather than from retrieval competition alone.

### 1.4. The Present Work

While prior research has primarily attributed memory interference to retrieval competition, emerging evidence suggests that interference can also originate from encoding-based reorganization of overlapping memories. As reviewed above, reactivation of prior associations during new learning can lead to either integration or differentiation, depending on the strength and nature of hippocampal reactivation. These encoding-based processes determine how overlapping traces are represented—either as cohesive, shared networks or as distinct, segregated memories—thereby shaping the degree of cue-driven competition that arises at retrieval. From this integrated perspective, interference reflects not only the outcome of competition among coactivated traces but also the representational structure established during encoding that gives rise to such competition.

The present study aims to disentangle these interacting mechanisms by examining associative and source memory performance in a paradigm that manipulates cue overlap while minimizing direct retrieval competition. Specifically, participants studied two lists of picture–word pairs (A-B and A-C), where overlapping cues (A) were paired with different associates across lists, and non-overlapping pairs (E-F, G-H) served as controls. Associative recognition was assessed using a two-alternative forced-choice (2AFC) task designed to minimize retrieval competition. In this task, each target was paired with a same-list distractor, thereby reducing direct cue-based competition between overlapping associations (e.g., A-B and A-C) during retrieval. This approach allows performance to more directly reflect the strength and distinctiveness of the encoded associations, while still acknowledging that some residual retrieval overlap may occur. Following recognition, participants completed a source memory judgment to indicate the list of origin for correctly recognized pairs. Together, these tasks provide a process-level framework for examining how encoding-related reactivation and retrieval-related competition jointly contribute to associative interference.

#### Research Questions and Hypotheses

Presence and Asymmetry of Interference. If interference arises solely from retrieval competition, both retroactive (RI) and proactive (PI) interference should be attenuated under non-competitive test conditions. However, if interference also stems from encoding-based processes, differences between cue-overlap and non-overlap pairs should still emerge. Specifically, we predict that A-B accuracy (List 1) may decline following A-C learning (RI), or that A-C accuracy (List 2) may decline due to prior A-B learning (PI). Asymmetric effects—where RI and PI differ in magnitude—would support the view that encoding dynamics, such as integration or differentiation, play a selective role in shaping interference.

Source Memory as a Diagnostic of Encoding Reorganization. If integration occurs during A-C encoding, the shared cue representation should blur the boundary between lists, reducing source discrimination accuracy and prolonging response times (RTs). In contrast, differentiation should yield the opposite pattern—sharper source distinctions and faster RTs for cue-overlap pairs. Thus, source memory performance provides a behavioral marker of how overlapping traces were reorganized at encoding.

Latency as an Index of Retrieval Dynamics. Beyond accuracy, response times in associative recognition and source judgments offer sensitive measures of retrieval fluency and competition. Longer RTs for overlapping pairs, even when accuracy remains comparable, would indicate that latent competition or reduced distinctiveness delays evidence accumulation during decision-making. Such effects would reveal the functional consequences of encoding-based representational overlap during retrieval.

By combining accuracy and latency measures across associative and source memory tasks, the present study provides a comprehensive behavioral test of the encoding–retrieval interaction framework. This design isolates how cue overlap and reactivation at encoding shape later retrieval dynamics, clarifying whether interference effects in associative memory can emerge independently of retrieval competition and revealing how representational reorganization contributes to both proactive and retroactive interference.

## 2. Method

Experiments were approved by the University’s Institutional Review Board. All participants read and signed an informed consent form and were compensated appropriately upon completion. The experiment was approved by the Ethics Committee of the host institution.

### 2.1. Design

We used a 2 (List: List 1 vs. List 2) × 2 (Pair type: Cue overlap vs. Non-overlap) repeated-measure design. Participants completed two lists of memory tasks: an initial list followed by a second list. Both lists contained both cue overlap and non-overlap associations to measure interference effects. The dependent variables were accuracy and reaction time for associative recognition, and accuracy and reaction time for list source judgments, specifically for associative recognition pairs that were successfully identified.

### 2.2. Participants

Based on prior research and an a priori power analysis (α = 0.05, power = 0.80, two-tailed), a sample size of 78 participants was determined to be sufficient to detect a medium effect size (*f* = 0.25) ([5]). Seventy-eight undergraduate students (25 men, 53 women; *M* = 21.94 years, *SD* = 2.40) were recruited for the study. All participants were right-handed and reported normal or corrected-to-normal vision. Each participant was tested individually in a quiet, controlled laboratory environment.

### 2.3. Materials

The experimental materials included both verbs and images, selected and arranged to create conditions of cue overlap and non-overlap in list 1 and list 2. A total of 100 Chinese verbs were selected from the Chinese Affective Words System (CAWS) ([36]). Of these, 4 were reserved for practice trials. The remaining 96 verbs were divided into four sets of 24, matched on emotional valence and familiarity. Mean valence ratings (*M* ± *SD*) were: Group 1 (4.92 ± 1.59), Group 2 (4.90 ± 1.50), Group 3 (4.62 ± 1.54), and Group 4 (5.11 ± 1.50). Familiarity ratings were comparable: Group 1 (5.09 ± 1.95), Group 2 (5.07 ± 1.96), Group 3 (5.10 ± 1.93), and Group 4 (5.06 ± 2.07). 76 images were selected from the Bank of Standardized Stimuli (BOSS) ([3]), with four images reserved for practice trials. The remaining 72 experimental images were evenly divided into two categories: biotic (e.g., mammals, birds, insects; *n* = 36) and abiotic (e.g., furniture, clothing, tools, buildings; *n* = 36). All images were highly familiar to participants. The 36 biotic and 36 abiotic images were further randomized into three groups, each containing 12 biotic and 12 abiotic images. The groups were matched on both familiarity (*M* ± *SD*: Group 1 = 4.43 ± 0.78, Group 2 = 4.50 ± 0.80, Group 3 = 4.51 ± 0.75) and visual complexity (Group 1 = 2.62 ± 1.27, Group 2 = 2.56 ± 1.24, Group 3 = 2.61 ± 1.25), with no significant differences across groups (all *p*s > 0.05).

One set of 24 images was used in both List 1 and List 2. In List 1, each image was paired with a unique verb (A-B); in List 2, the same image was paired with a different, unrelated verb (A-C). These formed the cue-overlap pairs (A-B, A-C), where the image served as the repeated retrieval cue, and B and C were pre-assigned with low semantic relatedness. To verify low semantic overlap between co-paired verbs (B and C), 24 participants rated the association strength of the A-B, A-C word pairs on a 7-point Likert scale (1 = not at all associated, 7 = very strongly associated). The average rating was low (*M* = 2.43, *SD* = 0.48), confirming minimal semantic association.

The remaining 48 images (two sets) were used only once: one set appeared in List 1 and the other in List 2, paired with different sets of verbs (E-F or G-H). These formed the non-overlap pairs in list1 and list2, with no shared cues across lists.

### 2.4. Procedure

The experiment was programmed in E-Prime 3.0 (Psychology Software Tools) and presented on a 16-inch laptop with 1920 × 1080 resolution (60 Hz refresh rate). Participants responded via keyboard while maintaining a viewing distance of approximately 40 cm from the screen center. Images (500 × 500 pixels) were presented in the upper center of the screen, with words displayed directly below each image. Participants first completed a 4-trial practice block using stimuli not included in the main experiment. The formal experiment consisted of five sequential phases:

List 1 learning phase. Participants studied list 1 word–image pairs. Half of these were cue-overlap pairs (A-B type)—images that would later be repeated in List 2 with a different verb. The other half were non-overlap pairs (E-F type)—images and words that appeared only in List 1. Each trial began with a 500 ms fixation cross, followed by a 3000 ms presentation of a word–image pair. Pairs were presented in a randomized order.

List 2 learning phase. During the List 2 learning phase, participants first completed a 60 s Chinese Cancellation Task to prevent rehearsal of the List 1 materials. They then studied a new set of word–image pairs. For half of the trials, images from the cue-overlap set in List 1 were repeated and paired with new, unrelated words (A-C pairs). The remaining trials consisted of entirely new word–image pairs (G-H pairs), forming the non-overlap condition for List 2.

Test Phase. Participants completed two tasks. In the Associative Recognition Task, participants were presented with an image and two previously studied words (e.g., Investigate vs. march) and were asked to select the word that had originally been paired with the image. Both the target and the foil items were drawn from the same list (either List 1 or List 2), ensuring that associative recognition for A-B pairs (List 1) and for A-C pairs (List 2) was tested independently. After selecting the correct word, participants indicated whether the chosen word–image pair had been studied in List 1 or List 2. The order of test pairs from List 1 and List 2 was randomized across trials. A schematic illustration of the entire procedure is shown in Figure 1.

## 3. Results

Data analysis was conducted using R version 4.5.0 ([26]). A two-way repeated-measures analysis of variance (ANOVA) was performed using the *afex* package (version 1.3-0; [33]) to examine the effects of pair type (cue-overlap vs. non-overlap) and list (List 1 vs. List 2) on associative recognition accuracy, with a significance level set at α = 0.05. Trials with reaction times (RTs) exceeding 5000 ms (2.36% of total trials) were excluded because the experimental design automatically advanced to the next trial after 5000 ms, indicating no response. Additionally, trials with recognition RTs < 200 ms or > 3*SD*, and source judgment RTs < 100 ms or > 3*SD*, were removed to ensure data quality.

To complement the frequentist analysis and provide a more nuanced assessment of the strength of evidence, Bayesian analyses were also conducted using the BayesFactor package in R (version 1.0.0; [18]). This approach allows quantifying the relative evidence for the alternative versus the null hypothesis through Bayes factors, offering a continuous measure of evidential strength rather than relying solely on threshold-based significance testing. The inclusion of Bayesian inference enhances the interpretability and robustness of the findings, aligning with best practices in contemporary psychological research ([34], [35]).

All data, analysis scripts, and materials are publicly available at the Open Science Framework: https://osf.io/pfgca/ (accessed on 21 October 2025).

### 3.1. Associative Recognition Accuracy

Descriptive statistics for associative recognition accuracy are presented in Table 1. A 2 (List: List 1 vs. List 2) × 2 (Pair Type: Cue-Overlap vs. Non-Overlap) repeated-measures ANOVA revealed a significant main effect of list, *F*(1, 77) = 13.31, *p* < 0.001, partial *η*^2^ = 0.15, with higher accuracy observed in List 1 (*M* = 0.78, *SD* = 0.14) than in List 2 (*M* = 0.75, *SD* = 0.14). A significant main effect of pair type was also found, *F*(1, 77) = 5.01, *p* = 0.028, partial *η*^2^ = 0.06, indicating higher accuracy for non-overlap pairs (*M* = 0.78, *SD* = 0.15) than for cue-overlap pairs(*M* = 0.75, *SD* = 0.13). The interaction between list and pair type was also significant, *F*(1, 77) = 9.28, *p* = 0.003, partial *η*^2^ = 0.11 (see Figure 2).

To explore this interaction, simple effects analyses were conducted using the *emmeans* package (version 1.10.0; [13]). For List 1, accuracy was significantly higher for non-overlap pairs (*M* = 0.81, *SD* = 0.13) compared to cue-overlap pairs (*M* = 0.75, *SD* = 0.15), *t*(77) = −4.45, *p* < 0.001, Cohen’s *d* = −0.50. In contrast, for List 2, there was no significant difference between cue-overlap (*M* = 0.75, *SD* = 0.11) and non-overlap pairs (*M* = 0.74, *SD* = 0.17), *t*(77) = 0.68, *p* = 0.50, Cohen’s *d* = 0.08. All *p*-values are two-tailed and Bonferroni-adjusted where applicable.

To quantify the strength of evidence for each effect, a Bayesian repeated-measures ANOVA was conducted with list and pair type as within-subject factors (default Cauchy prior width r = 0.707).

The results provided strong evidence for a main effect of list (BF_10_ = 20.48), but only anecdotal evidence for a main effect of pair type (BF_10_ = 1.03).

The inclusion of the list and pair type interaction did not improve model fit (BF_10_ = 1.01), indicating that the evidence for an interaction was inconclusive.

Follow-up Bayesian paired-sample *t*-tests showed strong evidence for a difference between cue-overlap and non-overlap pairs in List 1 (BF_10_ = 643.0), but moderate evidence for no difference in List 2 (BF_01_ = 6.43).

These results suggest that while cue overlap influenced recognition accuracy in List 1, the effect did not generalize to List 2.

### 3.2. Associative Recognition Reaction Times

To complement the accuracy analysis, recognition reaction times (RTs) were analyzed to examine potential interference effects that may not be evident from accuracy alone.

Importantly, RTs from both correct and incorrect recognition trials were included in the analysis. Reporting RTs for all response outcomes is critical for several methodological and theoretical reasons. First, it rules out a potential speed–accuracy trade-off: if shorter RTs for correct trials merely reflected participants’ tendency to respond quickly at the expense of accuracy, error trials would be expected to show even faster responses. Instead, the data revealed the opposite pattern—errors were slower than correct responses—demonstrating that RT differences genuinely reflect variations in memory strength rather than strategic speed biases.

Second, analyzing RTs for incorrect trials provides insight into the nature of retrieval processes. Longer latencies on error trials indicate that incorrect responses stem not from impulsive guessing, but from more effortful and ultimately unsuccessful retrieval attempts. Third, these findings align with decision models of memory (e.g., diffusion or evidence accumulation models), which predict that decisions based on weak or ambiguous mnemonic evidence take longer to reach, regardless of accuracy. Finally, including both correct and incorrect responses ensures full data transparency and strengthens the interpretability of the results.

RTs were subjected to a 2 (List: List 1 vs. List 2) × 2 (Pair Type: Cue-Overlap vs. Non-Overlap) × 2 (Recognition Accuracy: Correct vs. Incorrect) repeated-measures ANOVA using the *afex* package (version 1.3-0; [33]). Due to incomplete data, 7 participants were excluded, resulting in a final sample of 71 participants. Descriptive statistics are presented in Table 1, and patterns are illustrated in Figure 3.

The analysis revealed a significant main effect of list, *F*(1, 70) = 6.99, *p* = 0.010, partial *η*^2^ = 0.09, with shorter RTs in List 1 (*M* = 2491, *SD* = 353.7) than in List 2 (*M* = 2574, *SD* = 346.9). There was also a significant main effect of pair type, *F*(1, 70) = 15.12, *p* < 0.001, partial *η*^2^ = 0.18, with longer RTs for cue-overlap pairs (*M* = 2592, *SD* = 357.6) than non-overlap pairs (*M* = 2473, *SD* = 341.5). A highly significant main effect of recognition accuracy was found, *F*(1, 70) = 82.41, *p* < 0.001, partial *η*^2^ = 0.54, reflecting faster responses for correct (*M* = 2367, *SD* = 273.3) versus incorrect recognition (*M* = 2698, *SD* = 423.8).

A significant interaction emerged between list and pair type, *F*(1, 70) = 10.72, *p* = 0.002, partial *η*^2^ = 0.13. No significant interactions were observed between recognition accuracy and the other factors, all *p*s > 0.05. Follow-up comparisons indicated that for List 1, the RT difference between cue-overlap (*M* = 2513, *SD* = 383.7) and non-overlap pairs (*M* = 2469, *SD* = 368.2) was not significant, *t*(70) = 1.21, *p* = 0.231, Cohen’s *d* = 0.12. For List 2, RTs were significantly longer for cue-overlap (*M* = 2671, *SD* = 380.4) than for non-overlap pairs (*M* = 2477, *SD* = 369.7), *t*(70) = 4.86, *p* < 0.001, Cohen’s *d* = 0.51.

To further examine retrieval dynamics independent of recognition errors, a Bayesian repeated-measures ANOVA was conducted on reaction times (RTs) from correct recognition trials only, with list (List 1 vs. List 2) and pair type (Cue-Overlap vs. Non-Overlap) as within-subject factors (subject as a random factor; default Cauchy prior width r = 0.707).

Descriptive statistics indicated that mean RTs were slightly faster for non-overlap pairs (List 1: *M* = 2320 ms, *SD* = 347; List 2: *M* = 2273 ms, *SD* = 354) than for cue-overlap pairs (List 1: *M* = 2340 ms, *SD* = 363; List 2: *M* = 2513 ms, *SD* = 428).

At the model level, the Bayesian ANOVA provided anecdotal evidence for a main effect of list (BF_10_ = 2.04) and strong evidence for a main effect of pair type (BF_10_ = 3.59 × 10^4^), suggesting that responses to cue-overlap pairs were generally slower than to non-overlap pairs. The incremental evidence for including the list and pair type interaction beyond the main-effects model was anecdotal (BF_10_ = 1.01), indicating that the interaction effect was not strongly supported.

Follow-up Bayesian paired-sample *t*-tests were conducted within each list. For List 1, the Bayes factor favored the null model (BF_10_ = 0.16, BF_01_ = 6.44), providing moderate evidence for no RT difference between cue-overlap and non-overlap pairs. For List 2, however, the Bayes factor provided decisive evidence for a difference (BF_10_ = 9.01 × 10^6^), with non-overlap pairs eliciting faster responses than cue-overlap pairs.

Together, these findings indicate that, when recognition responses were correct, slower RTs for cue-overlap pairs primarily emerged in the second list, consistent with greater proactive interference during later encoding phases. The absence of a difference in List 1 suggests that retroactive interference was minimal when items were first encoded.

### 3.3. Source Judgement Accuracy

To examine source memory, we analyzed the proportion of correct source judgments for successfully recognized associations. A 2 (Pair Type: Cue-Overlap vs. Non-Overlap) × 2 (List: List 1 vs. List 2) repeated-measures ANOVA revealed a significant main effect of List, *F*(1, 77) = 4.79, *p* = 0.032, partial *η*^2^ = 0.06, with higher source accuracy in List 1 (*M* = 0.74, *SD* = 0.14) than in List 2 (*M* = 0.70, *SD* = 0.15). The main effect of pair type was not significant, *F*(1, 77) = 0.77, *p* = 0.383, partial *η*^2^ = 0.01, nor was the interaction, *F*(1, 77) = 0.07, *p* = 0.790, partial *η*^2^ < 0.001. These findings suggest that cue overlap did not significantly affect source memory, although source accuracy was generally better for List 1. Descriptive statistics are shown in Table 2 and Figure 4.

To further assess source recognition performance beyond frequentist inference, a Bayesian repeated-measures ANOVA was conducted on accuracy data, with list (List 1 vs. List 2) and pair type (Cue-Overlap vs. Non-Overlap) as within-subject factors (subjects as a random factor; default Cauchy prior width r = 0.707).

The analysis provided moderate evidence for a main effect of list (BF_10_ = 4.06), indicating that overall source recognition accuracy was slightly higher for List 1 (*M* = 0.75, *SD* = 0.14) than for List 2 (*M* = 0.70, *SD* = 0.15). In contrast, there was moderate evidence for the null hypothesis regarding the main effect of pair type (BF_10_ = 0.16, BF_01_ = 6.25), suggesting that accuracy did not differ reliably between cue-overlap and non-overlap pairs. The inclusion of the list and pair type interaction yielded anecdotal evidence (BF_10_ = 1.01) in favor of an interaction, indicating that the data were essentially insensitive to this effect. To further examine potential list-specific effects, Bayesian paired-sample *t*-tests were performed within each list. For both List 1 (BF_10_ = 0.17, BF_01_ = 5.74) and List 2 (BF_10_ = 0.13, BF_01_ = 7.48), the Bayes factors provided moderate evidence supporting the null model, indicating that source recognition accuracy did not differ between cue-overlap and non-overlap pairs in either list.

Taken together, these Bayesian results suggest that while participants’ overall source memory was slightly better for items from the first list, pair-type overlap did not systematically influence source recognition accuracy, providing no compelling evidence for either proactive or retroactive interference in source attribution performance.

### 3.4. Source Judgement Reaction Times

In addition to accuracy analyses, response times (RTs) during the source judgment task were also examined to provide a more comprehensive understanding of the retrieval process. Reporting RTs alongside accuracy serves two purposes. First, it allows us to evaluate whether differences in performance are driven by memory processes rather than by a speed–accuracy trade-off. Second, it offers insights into the cognitive dynamics underlying correct and incorrect source judgments.

A 2 (List) × 2 (Pair Type) × 2 (Source Accuracy: Correct, Incorrect) repeated-measures ANOVA was conducted to examine mean response times during the source judgment task. Six participants were excluded due to missing data, yielding a final sample of 72 participants. Descriptive statistics are reported in Table 2 and Figure 5.

There was a significant main effect of source accuracy, *F*(1, 71) = 39.17, *p* < 0.001, partial *η*^2^ = 0.36, indicating faster RTs for correct judgments (*M* = 616.5, *SD* = 374.8) than incorrect judgments (*M* = 700.0, *SD* = 425.8). This pattern rules out a speed–accuracy trade-off: if errors reflected hasty or impulsive responding, error RTs would have been shorter. Instead, the opposite pattern was observed—errors were slower, suggesting that incorrect responses arose from more effortful and less fluent retrieval attempts. This RT asymmetry provides an important window into the cognitive mechanism of source memory. Correct judgments likely reflect the retrieval of strong and distinctive contextual representations, supporting fluent and confident decisions. In contrast, incorrect judgments are presumably based on weak, ambiguous, or competing memory traces, resulting in prolonged decision times as participants search for or evaluate insufficient memory evidence. This interpretation aligns with evidence-accumulation models of decision making, which propose that responses based on degraded evidence take longer to reach a decision threshold, even when the final decision is incorrect.

The main effect of pair type was marginal, *F*(1, 71) = 3.36, *p* = 0.071, partial *η*^2^ = 0.05, suggesting slightly faster RTs for cue-overlap (*M* = 645.5, *SD* = 392.8) compared to non-overlap pairs (*M* = 671.0, *SD* = 407.8). The main effect of the list was not significant, *F*(1, 71) = 0.02, *p* = 0.880, partial *η*^2^ < 0.01.

Significant two-way interactions were found between list and pair type, *F*(1, 71) = 8.96, *p* = 0.004, partial *η*^2^ = 0.11, and between list and source accuracy, *F*(1, 71) = 8.09, *p* = 0.006, partial *η*^2^ = 0.10. A significant three-way interaction was also observed among list, pair type, and source accuracy, *F*(1, 71) = 4.85, *p* = 0.031, partial *η*^2^ = 0.06. To follow up on the three-way interaction, we first examined the interaction between list and pair type separately for correct and incorrect source judgments. When source judgments were incorrect, the interaction between list and pair type was significant, *F*(1, 71) = 8.68, *p* = 0.005. Simple effects analyses revealed that in List 1, RTs were significantly slower for non-overlap pairs (*M* = 764, *SD* = 443.0) than for cue-overlap pairs (*M* = 674, *SD* = 406.0), *t*(71) = −2.46, *p* = 0.017, Cohen’s *d* = −0.29. In contrast, in List 2, the difference between cue-overlap (*M* = 710, *SD* = 427.0) and non-overlap pairs (*M* = 652, *SD* = 427.0) was not significant, *t*(71) = 1.65, *p* = 0.104. For correct source judgments, the interaction between list and pair type was not significant, *F*(1, 71) = 1.02, *p* = 0.316. However, a significant simple effect of pair type was observed in List 1, with cue-overlap pairs eliciting faster responses (*M* = 572, *SD* = 344) than non-overlap pairs (*M* = 619, *SD* = 381), *t*(71) = −2.55, *p* = 0.013, Cohen’s *d* = −0.30. In List 2, the difference was non-significant, *t*(71) = −1.30, *p* = 0.198, with mean RTs of 626 (*SD* = 394.0) for cue-overlap and 649 (*SD* = 380) for non-overlap pairs.

To further verify the results of the frequentist repeated-measures ANOVA, a Bayesian repeated-measures ANOVA was conducted on the mean source-recognition reaction time (RT) for correctly recognized trials, with List (List 1 vs. List 2) and Pair Type (Cue-overlap vs. Non-overlap) as within-subject factors. Default Cauchy priors were used (r = 0.707).

Across participants, mean RTs were 565 ms (*SD* = 170) for cue-overlap and 611 ms (*SD* = 184) for non-overlap pairs in List 1, and 619 ms (*SD* = 203) and 639 ms (*SD* = 184, respectively, in List 2. Overall, recognition decisions were faster for cue-overlap than for non-overlap pairs, particularly in List 1.

The Bayesian ANOVA revealed strong evidence for a main effect of List (BF_10_ = 41.30), indicating that mean RTs differed reliably between the two lists. There was moderate evidence for a main effect of Pair Type (BF_10_ = 4.55), suggesting that RTs were overall shorter for cue-overlap than for non-overlap pairs. The inclusion of the List and Pair Type interaction only provided anecdotal evidence (BF_10_ = 1.01) beyond the main-effects model, implying that the evidence for an interaction effect is weak and inconclusive.

To further clarify the direction and magnitude of the effects within each list, Bayesian paired-samples *t* tests compared the two pair types separately for each list.

In List 1, there was moderate evidence for a difference between cue-overlap and non-overlap pairs (BF_10_ = 3.25), with faster responses for cue-overlap pairs. In List 2, however, the Bayes factor (BF_10_ = 0.25) provided moderate evidence for the null hypothesis, indicating no meaningful difference between the two pair types in that list.

These Bayesian findings were largely consistent with the frequentist ANOVA in revealing reliable main effects of both List and Pair Type. However, unlike the frequentist analysis, which detected a statistically significant List and Type interaction, the Bayesian results provided only anecdotal evidence for this interaction (BF_10_ = 1.01), indicating that the strength of evidence for the interaction is weak.

Importantly, the pattern of reaction times revealed that source judgments for cue-overlap pairs were faster than those for non-overlap pairs, particularly in the first list. This result deviates from the initial prediction that cue overlap would impair source discrimination due to increased mnemonic similarity or integration. Instead, the shorter RTs for cue-overlap pairs suggest that the retrieval process for these associations was more fluent and efficient. This pattern implies that the overlapping cue context may have facilitated the differentiation of memory traces rather than producing interference, reflecting a stronger and more distinctive encoding of the source information associated with shared cues.

## 4. Discussion

### 4.1. Differential Manifestations of Retroactive and Proactive Interference

The present study revealed distinct patterns for retroactive (RI) and proactive interference (PI) in associative recognition and source memory tasks. In List 1, cue-overlap pairs (A-B) showed reduced recognition accuracy relative to non-overlapping pairs (E-F), whereas their reaction times did not differ significantly. This selective accuracy impairment without a corresponding latency cost is consistent with retroactive interference, suggesting that subsequent A-C learning disrupted access to previously encoded A-B associations. In contrast, in List 2, recognition accuracy did not differ between cue-overlap (A-C) and non-overlap (G-H) pairs, yet reaction times were markedly slower for overlapping associations. This latency cost in the absence of accuracy differences indicates proactive interference, reflecting increased retrieval difficulty caused by residual activation of earlier A-B traces.

Together, these findings demonstrate that RI and PI, though both arising from overlapping associative structures, manifest differently: RI primarily weakens memory strength or accessibility, while PI primarily slows retrieval fluency through competition between coactivated traces. These behavioral dissociations support the view that interference reflects multiple underlying mechanisms, which may dominate at different learning stages or depend on the representational organization of memory traces.

### 4.2. Mechanistic Differences Between RI and PI: Encoding and Retrieval Contributions

The asymmetrical expression of RI and PI observed here aligns with the notion that the two forms of interference may stem from distinct phases of memory processing ([23]; [25]).

Retroactive interference (RI) appears to arise mainly from (re)encoding-based reorganization: when List 2 cue-overlap pairs (A-C) are encoded, partial reactivation of List 1 cue-overlap pairs (A-B) traces can trigger representational integration or differentiation within the hippocampal–mPFC network ([4]; [12]; [30]). In cases where reactivation is strong, pattern completion promotes integration between A-B and A-C, producing blended representations that reduce the distinctiveness of the original association and impair later recognition—a hallmark of RI.

In contrast, proactive interference (PI) likely reflects retrieval-based competition: when participants attempt to retrieve A-C associations, residual activation of previously learned A-B pairs coactivates competing responses, delaying selection of the correct target and thereby prolonging reaction times ([1]; [16]). The presence of a strong PI-related latency cost, even under non-competitive test conditions, suggests that the competition originates from cue-driven coactivation of overlapping representations, rather than from explicit decision conflict at test.

Thus, RI and PI differ not only in behavioral manifestation but also in their underlying mechanisms—RI reflects representational interference formed during encoding, whereas PI reflects retrieval interference among coactivated traces. These complementary processes jointly contribute to the overall pattern of associative interference observed in episodic memory.

### 4.3. Encoding–Retrieval Interaction in Associative Interference

The combined results from associative recognition and source memory tasks support an integrated framework in which encoding-based reactivation and retrieval-based competition interact dynamically to shape memory performance.

At encoding, overlapping associations trigger reactivation of prior traces, leading to either integration (shared, overlapping representations) or differentiation (distinct, segregated representations). These representational outcomes determine the degree of overlap that later governs retrieval competition. When integration predominates, cue overlap amplifies associative similarity, increasing retrieval competition and thus producing RI or PI effects. Conversely, when differentiation occurs, overlapping traces become more distinct, reducing interference and potentially facilitating faster, more accurate retrieval.

The current finding that source judgments were faster for cue-overlap pairs, particularly in List 1, provides behavioral evidence consistent with differentiation. Rather than producing confusion between contexts, shared cues appeared to enhance the distinctiveness of their associated sources, enabling more fluent retrieval of contextual information. This pattern suggests that, under certain encoding conditions, reactivation may promote representational sharpening rather than blending, reducing cross-list interference while strengthening contextual discrimination.

Together, these results demonstrate that encoding-based and retrieval-based mechanisms are not independent but mutually constraining. Encoding determines the structural similarity among memory traces through reactivation-driven reorganization, whereas retrieval reveals the functional consequences of that structure through competitive access dynamics. The interaction between these processes explains why RI and PI can vary asymmetrically across learning lists and why interference effects persist even in non-competitive testing contexts.

This study extends traditional retrieval-based accounts of interference (e.g., [1]; [27]) by showing that interference can arise even when overt retrieval competition is minimized, underscoring the importance of encoding dynamics in shaping later memory accessibility. By integrating the principles of pattern reactivation and reorganization from recent neurocognitive models ([30]; [32]) with the classic retrieval competition framework, the current findings highlight that interference emerges from the interaction between how memories are structured at encoding and how they are accessed at retrieval.

In this view, encoding and retrieval form a continuous, interactive loop: encoding determines representational similarity, which governs retrieval competition, while retrieval outcomes feed back to influence future encoding through selective reactivation. Understanding interference, therefore, requires a dynamic, cross-stage perspective—one that integrates both representational and functional dimensions of memory. This framework not only reconciles prior discrepancies between RI and PI findings but also provides a process-level explanation for how overlapping experiences can produce either competition or facilitation, depending on the nature of encoding reactivation.

### 4.4. Limitations and Future Directions

While the present study offers important insights into the asymmetric mechanisms of retroactive and proactive interference and the central role of encoding processes, several limitations warrant consideration. First, the use of weakly associated word–picture pairs and brief encoding durations may have constrained the engagement of recollection-based retrieval, potentially amplifying familiarity-driven recognition effects. Given that associative recognition often relies on recollection, future studies should systematically manipulate associative strength and encoding duration to disentangle the relative contributions of familiarity and recollection under varying memory demands ([39]; [38]).

Second, although the two-alternative forced-choice (2AFC) paradigm employed here effectively minimized retrieval competition, it may not fully capture the complexity of naturalistic memory interference, where multiple associative traces may simultaneously compete during recall. Incorporating complementary paradigms—such as free recall, cued recall, or retrieval-induced forgetting tasks—could provide a richer characterization of how encoding- and retrieval-based mechanisms jointly shape interference effects ([1]; [9]).

Third, while we highlight the pivotal role of encoding dynamics (e.g., pattern separation and differentiation) in shaping interference, this study did not directly measure neural correlates of encoding processes, such as hippocampal activity patterns or neural similarity metrics. Future research could integrate neuroimaging or computational modeling approaches (e.g., CLS models, [22]) to test how encoding operations like pattern suppression or representational repulsion ([7]) mediate the trade-off between reduced interference and potential costs to recollection.

Finally, individual differences in encoding strategies, cognitive control, and working memory capacity may systematically modulate the degree of interference observed. Our findings suggest that adaptive encoding processes (e.g., differentiation) can either mitigate or exacerbate interference depending on individual strategy use. Future studies could examine these factors by including executive function measures or strategy assessments as covariates or moderators, which would deepen our understanding of variability in interference effects across individuals.

In summary, while this study advances the theoretical account of memory interference by emphasizing encoding as a critical locus of both retroactive and proactive effects, future research should adopt multi-method and multi-level approaches—combining behavioral paradigms, neural measures, and computational modeling—to further clarify the interplay between encoding and retrieval processes in shaping associative memory.

## 5. Conclusions

The present study demonstrates that associative interference can emerge even when retrieval competition is minimized, highlighting that the origins of memory interference extend beyond retrieval to include dynamic representational processes occurring during encoding. By combining a non-competitive recognition paradigm with source-memory measures, we revealed distinct behavioral signatures of retroactive and proactive interference—reflecting asymmetric contributions of encoding- and retrieval-related mechanisms.

These findings advance a process-interaction framework, in which encoding-based reactivation and reorganization shape the representational structure of overlapping memories, while retrieval competition reveals the functional consequences of that structure. From this perspective, memory interference is not a fixed outcome of cue competition but a dynamic consequence of how the memory system balances integration and differentiation across time.

By emphasizing the constructive and adaptive nature of encoding, the current work bridges classical retrieval-based models and contemporary representational accounts, offering a unified view of how similarity, reactivation, and competition jointly determine the accessibility of episodic associations. Future research combining behavioral, neural, and computational approaches will be crucial for capturing this interplay across levels of analysis and for refining theories of how the mind organizes, protects, and transforms its overlapping experiences.

## Figures and Tables

**Figure 1 behavsci-15-01459-f001:**
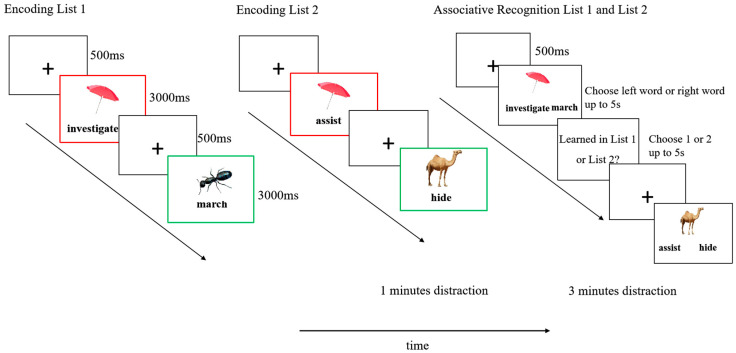
Experimental procedure. Participants studied two lists of word-picture pairs, each containing half cue overlap pairs (A-B or A-C condition, shown in red frames for illustrative purposes) and half non-overlap pairs (E-F or G-H condition, shown in green frames for illustrative purposes).

**Figure 2 behavsci-15-01459-f002:**
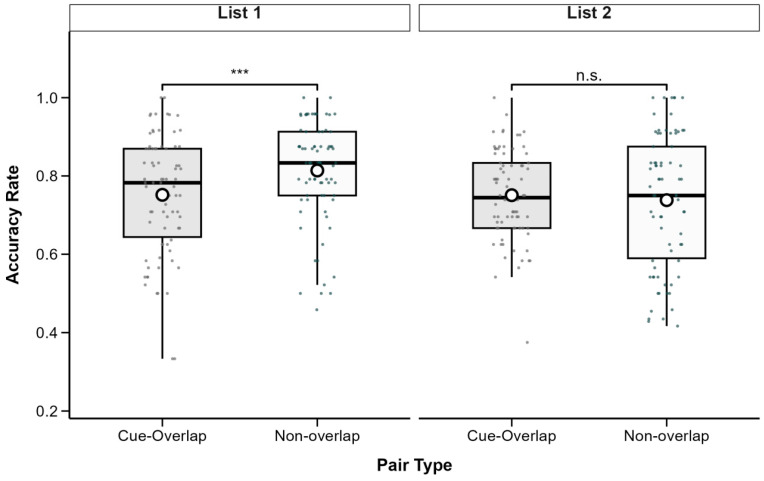
Associative recognition accuracy as a function of pair type and study list. Boxes show IQR (25th–75th percentiles), solid lines represent medians, whiskers extend to 1.5 × IQR, and white circles indicate condition means. Significant differences between cue-overlap and non-overlap pairs are marked with asterisks (*** *p* < 0.001); n.s. = not significant (*p* ≥ 0.05). Post hoc tests adjusted with Bonferroni correction.

**Figure 3 behavsci-15-01459-f003:**
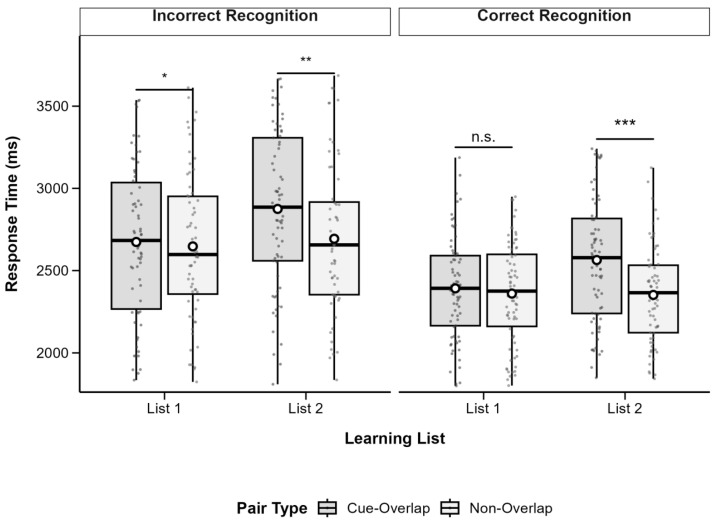
Boxplots of recognition response time by pair type and learning list, separated by recognition accuracy. Boxes show IQR (25th–75th percentiles), solid lines represent medians, whiskers extend to 1.5 × IQR, and white circles indicate condition means. Significant differences between cue-overlap and non-overlap pairs are marked with asterisks (* *p* < 0.05, ** *p* <0.01, *** *p* < 0.001); n.s. = not significant (*p* ≥ 0.05). Post hoc tests adjusted with Bonferroni correction.

**Figure 4 behavsci-15-01459-f004:**
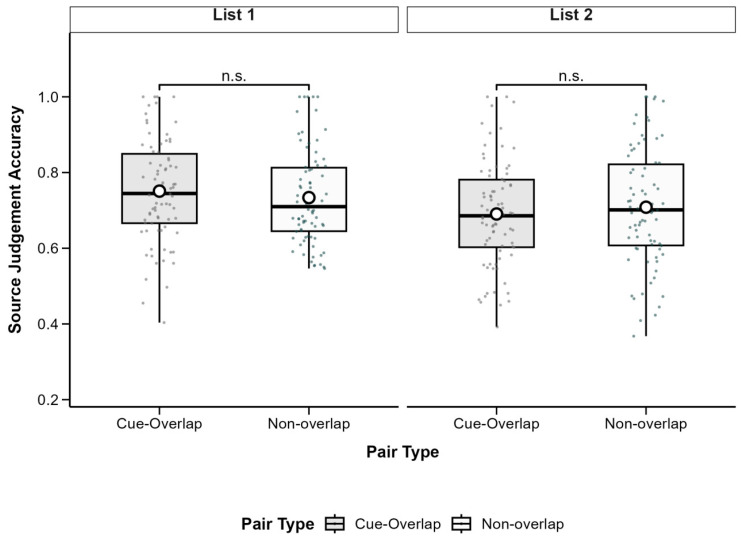
Boxplots of source judgement accuracy by pair type and learning list. Boxes show IQR (25th–75th percentiles), solid lines represent medians, whiskers extend to 1.5 × IQR, and white circles indicate condition means. Non-significant differences between conditions are marked as n.s (*p* ≥ 0.05). Post hoc tests adjusted with Bonferroni correction.

**Figure 5 behavsci-15-01459-f005:**
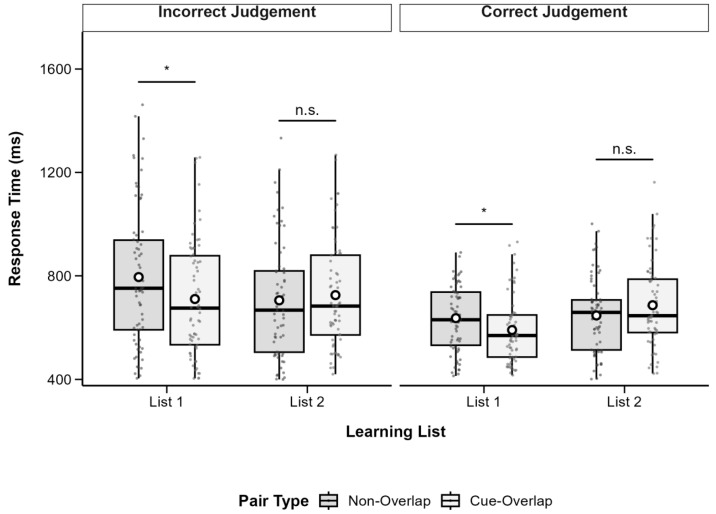
Boxplots of source judgement response time by pair type and learning list, separated by judgement accuracy. Boxes show IQR (25th–75th percentiles), solid lines represent medians, whiskers extend to 1.5 × IQR, and white circles indicate condition means. Significant differences between cue-overlap and non-overlap pairs are marked with asterisks (* *p* <0.05); n.s. = not significant (*p* ≥ 0.05). Post hoc tests adjusted with Bonferroni correction.

**Table 1 behavsci-15-01459-t001:** Mean (±SD) recognition accuracy and reaction times (ms) by pair type, list, and response accuracy.

List	Pair Type	Recognition Accuracy (*M* ± *SD*)	Correct Reaction Time (ms)	Incorrect Reaction Time (ms)
List 1	Cue-overlap	0.75 ± 0.15	2339.94 ± 363.04	2667.74 ± 647.66
	Non-overlap	0.81 ± 0.13	2319.70 ± 346.73	2631.99 ± 645.72
List 2	Cue-overlap	0.75 ± 0.11	2513.32 ± 427.81	2867.40 ± 572.10
	Non-overlap	0.74 ± 0.17	2273.00 ± 354.35	2668.13 ± 639.50

Note. RT = reaction time (in milliseconds). Accuracy is reported as proportion correct.

**Table 2 behavsci-15-01459-t002:** Mean Accuracy and Reaction Times for Source Judgments by Pair Type in List 1 and List 2.

List	Pair Type	Source Judgement Accuracy (*M* ± *SD*)	Correct Reaction Time (ms)	Incorrect Reaction Time (ms)
List 1	Cue-overlap	0.75 ± 0.15	561.45± 343.75	680.074 ± 406.29
	Non-overlap	0.73 ± 0.14	608.35 ± 381.49	750.08 ± 442.67
List 2	Cue-overlap	0.70 ± 0.15	620.62 ± 394.04	662.81 ± 427.18
	Non-overlap	0.69 ± 0.16	628.75 ± 379.66	667.05 ± 427.20

Note. RT = reaction time (in milliseconds). Accuracy is reported as proportion correct.

## Data Availability

Data or materials for the experiments are available at https://osf.io/pfgca/, and none of the experiments were preregistered. Code for the experiments is available at https://osf.io/pfgca/ (accessed on 21 October 2025).

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
