# Peer review of "Beyond Retrieval Competition: Asymmetric Effects of Retroactive and Proactive Interference in Associative Memory"

_behavsci, 2025, doi:10.3390/bs15111459_

Round 1

Reviewer 1 Report

Comments and Suggestions for Authors

Thank you for the opportunity to review your submission to Behavioral Science titled “Beyond Retrieval Competition: Asymmetric Effects of Retroactive and Proactive Interference in Associative Memory”. I enjoyed reading about the experiment you conducted that explored interference in episodic memory. Outcomes indicated that retroactive and proactive interference involved mechanisms during the encoding process, as opposed to processes in retrieval as many theories of inference suggest. This is a valuable and timely area of research. I have a few recommendations and comments regarding your manuscript, which I will discuss below.

Main Comments:

  1. One of my biggest concerns with the manuscript was the tone of the writing. Throughout the manuscript I found that the prose took on two voices. For example, much of the introduction was written in one voice, whereas the method and results seemed to take on another style and tone. As a reader, this made the article incredibly challenging to read. Specifically, parts of the introduction and discussion were so embroiled in jargon and acronyms that I was not able to follow along as a reader. In other parts that research was described in more common language that I could follow simply as the audience. In this way, I felt like there were times when I was reading two manuscripts spliced on top of each other. I recommend that the tone of the manuscript be unified into one. Specifically, I recommend adopting a more common tone, aiming to replace many of the acronyms with descriptive explanations. That is, instead of using A-B perhaps List 1 cue-overlap, and instead of E-F perhaps List 1 non-overlap pairs. As a reader this is easier to understand and would benefit the introduction and discussion of the manuscript.

  1. My second concern is related to scale of the claim emerging from this experiment. I am not convinced that the single experiment presented in this study is substantial enough to suggest that retrieval-based theories of interference are incorrect. The data of 78 participants is not sufficient to discredit the existing theories regarding retrieval-based interference. Along these lines, I have several potential paths forward.
    • When reading the introduction and method section, I found that I was not fully convinced that retrieval competition was accounted for in the design, making space for encoding and retention processes to be solely examined. Perhaps a more thorough explanation of why the procedures presented in the manuscript are sufficient would nullify these concerns.
    • Another option would be collecting additional experiments to better support the claim that encoding and retention are driving factors in interference. For example, the first set of follow-up experiments proposed in the limitations section were particularly compelling to me and would likely serve to strengthen the claim made here. An experiment that manipulates the associative strength and encoding duration to see how that moderated interference effects would be a strong addition here. Particularly since evidence for proactive interference in the current work is only in the form of reaction times. While the reaction time data does support the claim, it is less substantial than accuracy data and leaves area for further data to support this claim.
    • In lieu of these things, another option would be to soften the claim made in the discussion. The beginning of the discussion makes big claims about how encoding and retention play major roles in interference; however, the supporting data provides only one form of evidence for retroactive interference (accuracy data) and one form of evidence of proactive interference (RT data). Further, these data are not without outliers. Because the evidence is present but limited, it seems like a stretch to suggest a theoretical shift on this data alone. Perhaps I am missing the nuance in this particular area, but I think the evidence needs to become more compelling to make this claim. This could be through extension of the work, a better explanation of the present work, or a smaller claim.

  1. Was this work pre-registered? If not, I request that the authors add a statement to the paper confirming whether they have reported all measures, conditions, and data exclusions. The authors should, of course, add any additional text to ensure the statement is accurate. This is the standard reviewer disclosure request endorsed by the Center for Open Science. I include it in every review.

Minor Comments:

  1. Across the manuscript the font size and style changes randomly. Please correct this.
  2. I really enjoyed Figure 1, it was so useful in following the procedure!
  3. Some of the sub headers were oddly formatted. For example, on page 11, line 384, I think the top line is intended to be a sub header, but instead it is just in large font. Relatedly, section 2.4 in the method, the headers does not seem like a normal heading title and should be adjusted.

Reviewer 2 Report

Comments and Suggestions for Authors

Thank you for the opportunity to review the manuscript titled: “Beyond Retrieval Competition: Asymmetric Effects of Retroactive and Proactive Interference in Associative Memory”. First, I want to highlight that I am not a native speaker, and I apologize for any misunderstandings from my side.

The authors use within-subject mixed-list design to investigate effects of proactive (PI) and retroactive interference (RI) in associative memory. The results showed asymmetric effects, where RI manifests through decline in accuracy performance, and PI through increase in reaction time. The authors argue that such dissociation suggests different mechanisms for RI and PI. The authors also propose that by using their specific paradigm, they were able to disentangle encoding, and retrieval-based mechanisms, which might be underlying interference effects in associative memory. The authors propose that interference effects may arise not only through disruptions in retrieval mechanisms but may be the results of disruptions in the encoding stages.

I appreciate that the authors have shared data and codes for analyses.  I only have a few comments.

Major:

  • The non-significant outcome from NHST cannot distinguish between results that support the null hypothesis (no difference between the conditions) and results that are inconclusive. The authors should instead turn to Bayesian analyses to evaluate whether any non-significant results are in fact providing evidence that there was no difference between the conditions, or whether the results are inconclusive (e.g., because of too little data, see Wagenmakers et al., 2018a, 2018b)
  • Addressing the point above would also help the authors to avoid misleading interpretations, for example: “The main effect of pair type was marginal, F(1, 71) = 326 3.36, p = .071” – this seems like a misleading interpretation of the significance testing. Once something crosses our predetermined significance criteria, it is irrelevant how close or far from the criterion the p value is, the result is simply not statistically significant. Notably, just because it is not statistically significant, it does not provide us with information whether there was no difference between conditions, or whether the result is inconclusive (see my point above).

Wagenmakers, E.-J., Love, J., Marsman, M., Jamil, T., Ly, A., Verhagen, J., Selker, R., Gronau, Q. F., Dropmann, D., Boutin, B., Meerhoff, F., Knight, P., Raj, A., van Kesteren, E.-J., van Doorn, J., Šmíra, M., Epskamp, S., Etz, A., Matzke, D., … Morey, R. D. (2018). Bayesian inference for psychology. Part II: Example applications with JASP. Psychonomic Bulletin & Review, 25(1), 58–76. https://doi.org/10.3758/s13423-017-1323-7

Wagenmakers, E.-J., Marsman, M., Jamil, T., Ly, A., Verhagen, J., Love, J., Selker, R., Gronau, Q. F., Šmíra, M., Epskamp, S., Matzke, D., Rouder, J. N., & Morey, R. D. (2018). Bayesian inference for psychology. Part I: Theoretical advantages and practical ramifications. Psychonomic Bulletin & Review, 25(1), 35–57. https://doi.org/10.3758/s13423-017-1343-3

  • Even though the authors do provide a comprehensive theoretical framework, I think the explanation of how the expected outcomes may support the encoding-related mechanisms for memory interference should be clearer in the introduction. In general, sometimes the language seems over-complicated in the manuscript. I always think that the simpler and shorter the better. I believe that this manuscript readability would improve by simplifying the language and correcting grammatical errors (for an example, see below).

Minor:

  • Page 1, line 36: “Episodic memories are characterized by the composed of associative connections..:” – “characterized by” and “composed of” are redundant, use either one or the other. Rephrase to improve readability.
  • Read through to correct punctuation and small grammar errors (e.g., several sentences have a space before a period, sometimes a space is missing before parenthesis with reference or between two sentences – all should be corrected).
  • Reaction time in milliseconds is such a high number that it could be reported in seconds instead.

Round 2

Reviewer 1 Report

Comments and Suggestions for Authors

The authors did a commendable job editing the manuscript in response to my prior comments! My primary concerns regarding the tone of the manuscript and the scope of the claim have been thoroughly addressed. These edits improved the readability of the manuscript and provided much needed context to the introduction and results. I have only one minor comment that caught my eye while reading.

  • At the end of page 2 (around line 87), there is the following statement “Associative Unlearning posits that retroactive interference occurs during encoding, when learning cue-overlap associations in list 2 weakens prior similar associations in list 1 which independent of retrieval competition (Barnes & Underwood, 1959), especially when cue-outcome contingencies are violated (Mujezinović et al., 2024). This view, however, has primarily been supported in overlearning contexts, leaving its applicability to episodic memory less clear.”
    • Something about this sentence does not flow together. I think may be the transition between which and independent, but I do not know what this is trying to say.

Author Response

Comments 1: I have only one minor comment that caught my eye while reading.At the end of page 2 (around line 87), there is the following statement “Associative Unlearning posits that retroactive interference occurs during encoding, when learning cue-overlap associations in list 2 weakens prior similar associations in list 1 which independent of retrieval competition (Barnes & Underwood, 1959), especially when cue-outcome contingencies are violated (Mujezinović et al., 2024). This view, however, has primarily been supported in overlearning contexts, leaving its applicability to episodic memory less clear.”Something about this sentence does not flow together. I think may be the transition between which and independent, but I do not know what this is trying to say.

Response 1:We sincerely thank the reviewer for pointing out this issue and for the helpful observation regarding sentence flow and clarity. We have carefully revised the passage to improve its readability and precision. The revised version now reads as follows:

In contrast to retrieval-based accounts, the associative unlearning hypothesis proposes that retroactive interference arises during encoding rather than retrieval. Specifically, learning cue-overlapping associations in List 2 is thought to weaken previously acquired associations in List 1 (Barnes & Underwood, 1959). This weakening is particularly evident when cue–outcome contingencies are violated, disrupting the stability of existing associative links (Mujezinović et al., 2024). However, most evidence for associative unlearning has come from overlearning paradigms, and its relevance to typical episodic memory situations remains uncertain.

We appreciate the reviewer’s attentive reading and believe that this revision substantially improves the clarity and flow of the paragraph.